# OpenReview forum: "PRPO: Paragraph-level Policy Optimization for Vision-Language Deepfake Detection"
_ICLR.cc/2026/Conference — ICLR 2026 Conference Desk Rejected Submission_

### Official Review · Reviewer_jDoc · 2025-10-25

**Soundness:** 3
**Presentation:** 3
**Contribution:** 3
**Rating:** 6
**Confidence:** 3

**Summary:**

### Method
- The authors propose a new training dataset, an enhanced model, and a new RL algorithm for deepfake detection with MLLMs.
    - DF-R5, a new training set, which is collected by using off-the-shelf MLLMs such as Gemini.

    - An enhanced model, DX-LLaVA. The authors replace CLIP ViT with CLIP ConvNeXT to get pixel-level embeddings, enabling a finer focus on local image regions.

    - A new RL algorithm, PRPO. PRPO does not require ground truth labels, but utilize Visual Consistency Reward and Prediction Consistency Reward to improve the model performance, which can be applied at test time.

### Results
- PRPO achieves the highest reasoning score of 4.55/5.0.
- Ablation studies show the advantages of the proposed methods.

**Strengths:**

- This study improves deepfake detection in several aspects, including data, model architecture, and algorithms.
- The exploration and the proposed methods are intuitive and reasonable.
- PRPO significantly surpasses prior methods and achieves the highest reasoning score.

**Weaknesses:**

### Major
- Quality of the dataset.
    - DF-R5 is built on off-the-shelf MLLMs such as Gemini. I'm not sure whether this pipeline can ensure stable and high-quality data annotations. Does Gemini generate hallucination or wrong annotations?
    - Second, it seems that the upper bound of the data is limited by the used MLLMs.
    - Last, the evaluation such as the explanation quality evaluation also relies on MLLMs (GPT-4o here). I'm wondering if this leads to overestimated performance, because MLLMs generate training data and MLLMs evaluate. It is possible that these models will prefer an answer "similar" to their own.

- Generalization ability of DX-LLaVA.
    - As shown in Table 2, the performance for inter-domain is not satisfactory. I didn't see further discussion on this point, so I'm not sure about the generalization ability of DX-LLaVA across domains different from the training data

### Minor
- How is the performance of the classifer trained on the CLIP ConvNeXT features? Is it better than DX-LLaVA?
- The authors are recommended to put more examples in the main context or Appendix, to compare the quality of explanations generated by DX-LLaVA or other models.

- About PRPO
    - How do the authors separate paragraphs from the DX-LLaVA answers during training? A fixed-number of sentences is a paragraph? Or separate paragraphs based on meanings?
    - If we use ground truth labels to introduce accuracy rewards to PROP (like GRPO), will the performance be improved?

**Questions:**

Please see the Weaknesses section.

---

> ### Author Response · Authors · 2025-11-20
>
> We sincerely thank the reviewer for their constructive feedback and valuable suggestions. We address each concern below.
>
> ### Question 1: Dataset Quality - Hallucination and Annotations
>
> **Question:** DF-R5 is built on off-the-shelf MLLMs such as Gemini. I'm not sure whether this pipeline can ensure stable and high-quality data annotations. Does Gemini generate hallucination or wrong annotations? The upper bound of the data is limited by the used MLLMs.
>
> **Answer:** We appreciate this concern. Hallucination is indeed possible if a model is prompted to freely generate annotations. This is why DF-R5 was **not** built from direct Gemini outputs; instead, we designed a multi-stage pipeline specifically to reduce hallucination and ensure annotation stability. The generation pipeline incorporates several design choices:
>
> - **Structured scoring (Step 2 in Figure 2):** Instead of asking Gemini to produce free-form reasoning, we require it to score **74 predefined forensic features**. This greatly limits the space for hallucination and ensures consistency. Only features selected through this structured scoring are used per image.
>
> - **Multi-model validation:** The candidate features in Step 1 are obtained from **four different MLLMs**, not only Gemini. Consensus among multiple models substantially increases annotation reliability.
>
> - **Manual inspection to optimize prompts:** During dataset construction, we repeatedly sampled and manually inspected 100 random examples across all domains, refining prompts and filtering criteria until the outputs became stable and high-quality.
>
> - **Empirical validation through model performance:** State-of-the-art detectors fine-tuned on DF-R5 (e.g., SIDA, FakeShield, DE-FAKE) achieve strong cross-domain generalization (Table 4). Such gains would not be possible if the dataset contained significant hallucination or unreliable annotations.
>
> ---
>
> ### Question 2: Evaluation Circularity
>
> **Question:** The evaluation such as the explanation quality evaluation also relies on MLLMs (GPT-4o here). I'm wondering if this leads to overestimated performance, because MLLMs generate training data and MLLMs evaluate. It is possible that these models will prefer an answer "similar" to their own.
>
> **Answer:** This is a valid methodological concern. We mitigate potential evaluation bias through three complementary strategies:
>
> 1. **Discriminative, rubric-based evaluation (not preference matching).**
>    Instead of asking GPT-4o for open-ended preferences, we use a **structured scoring rubric** (Figure 9, Appendix A.4) covering classification accuracy, evidence grounding, reasoning quality, consistency, and uncertainty handling. This reduces stylistic bias because the model must assign numerical scores to explicit criteria. Moreover, GPT-4o (evaluator) and Gemini-2.5 (data generator) differ substantially in training corpus, architecture, and output style, reducing the risk of alignment-based favoritism.
>
> 2. **Standard practice in long-form reasoning research.**
>    Using strong MLLMs as scalable evaluators is now the accepted norm for long-form reasoning benchmarks. Our setup follows the evaluation methodology used in recent SOTA vision-language reasoning works [1][2].
>
> 3. **Human evaluation to verify alignment with GPT-4o.**
>    During the rebuttal period, we conducted a **human-evaluation study** with **8** participants, split into a *lower-expertise group* (4 researchers with general computer vision experience) and an expert group (4 researchers with specialized expertise in deepfake analysis). Each participant rated explanations from anonymous models on 5 randomly sampled images using the same rubric as GPT-4o.
>
> **Table: Human vs. GPT-4o Evaluation**
>
> | Group                   | Model      | CAC   | EGIA  | RQ    | CC    | CU    | Average |
> |-------------------------|------------|-------|-------|-------|-------|-------|---------|
> | **All Human Evaluators** | Gemini-2.5 | 4.225 | 4.350 | 4.275 | 3.950 | 4.300 | 4.220   |
> |                         | **PRPO**   | **5.00** | **4.625** | **4.475** | **4.575** | **4.475** | **4.630** |
> | **GPT-4o**          | Gemini-2.5 | 3.98  | 4.16  | 4.23  | 4.05  | 4.60  | 4.20    |
> |                         | **PRPO**   | **4.42** | **4.56** | **4.58** | **4.50** | **4.69** | **4.55** |
>
>
> The table shows that human judgments strongly align with GPT-4o's assessments, and both consistently rank PRPO highest across all evaluation criteria. The evaluation criteria (CAC, EGIA, RQ, CC, and CU) are presented in Section 4.3. This cross-validation confirms that GPT-4o provides reliable and unbiased evaluations of explanation quality.
>
> [1] Y Chen, Z Yan, G Cheng, K Zhao, S Lyu, B Wu. X2-dfd: A framework for explainable and extendable deepfake detection. NeurIPS, 2025.
>
> [2] Niki M Foteinopoulou, Enjie Ghorbel, and Djamila Aouada. A hitchhiker’s guide to fine-grained face forgery detection using common sense reasoning. NeurIPS, 2025.

---

> ### Author Response · Authors · 2025-11-20
>
> ---
>
> ### Question 3: Inter-domain Generalization
>
> **Question:** As shown in Table 2, the performance for inter-domain is not satisfactory. I didn't see further discussion on this point, so I'm not sure about the generalization ability of DX-LLaVA across domains different from the training data.
>
> **Answer:** We appreciate this concern, but we believe it stems from a misunderstanding of the experimental flow. Inter-domain generalization is the **core focus** of our work, and Table 2 serves only as the baseline that motivates the subsequent architectural and algorithmic improvements. The progression below summarizes how each component (ConvNeXT, DF-R5 fine-tuning, and PRPO) contributes to consistent and substantial cross-domain gains.
>
> **Table: Progressive Inter-Domain Generalization Improvements (%)**
>
> | Stage                            | Method                               | Acc        | F1         |
> |----------------------------------|--------------------------------------|------------|------------|
> | **1. Baseline (Table 2)**        | LLaVA (ViT)                          | 59.40      | 35.82      |
> | **2. Architectural Improvements** | + Classifier                         | 70.40      | 61.66      |
> |                                  | ConvNeXT + Classifier (**DX-LLaVA**) | 92.60      | 92.34      |
> |                                  | **Gain over LLaVA**                  | **+33.20** | **+56.52** |
> | **3. Test-Time Optimization**    | PRPO                                 | 95.80      | 95.88      |
> |                                  | **Gain over DX-LLaVA**               | +3.20      | +3.54      |
> | **Total Improvement**            | —                                    | **+36.40** | **+60.06** |
>
>
> ---
>
> ### Question 4: Classifier Performance
>
> **Question:** How is the performance of the classifier trained on the CLIP ConvNeXT features? Is it better than DX-LLaVA?
>
> **Answer:** We evaluated the performance of a binary classifier trained on frozen CLIP ConvNeXT features. The results are shown below:
>
> **Table: Performance of a Binary Classifier Trained on CLIP ConvNeXT Features**
>
> | Performance                  | Intra Domain F1 | Inter Domain F1 (DDIM) |
> |------------------------------|-----------------|------------------------|
> | Classifier    | 96.48           | 51.74                  |
> | DX-LLaVA  | **99.76**       | **91.54**              |
>
> The results indicate that the binary classifier achieves strong intra-domain performance (96.48% F1), indicating that CLIP ConvNeXT features are indeed effective for deepfake detection within known domains. However, its inter-domain generalization is limited (51.74% F1), substantially lower than DX-LLaVA's 91.54% F1. We find that the language model loss are critical for cross-domain robustness. It encourages the model to learn richer, more generalizable representations beyond what the frozen ConvNeXT features provide.
>
> ---
>
> ### Question 5: Qualitative Examples
>
> **Question:** The authors are recommended to put more examples in the main context or Appendix, to compare the quality of explanations generated by DX-LLaVA or other models.
>
> **Answer:** We fully agree with these suggestions. We have updated the Appendix with additional qualitative examples comparing explanations from Qwen2.5-VL-32B, Gemma-3-27B, Gemini-2.5, DX-LLaVA, and PRPO. Please refer to **Figures 13, 14, and 15** in Section A.7 of the Appendix.

---

> ### Author Response · Authors · 2025-11-20
>
> ---
>
> ### Question 6: Paragraph Separation in PRPO
>
> **Question:** How do the authors separate paragraphs from the DX-LLaVA answers during training? A fixed-number of sentences is a paragraph? Or separate paragraphs based on meanings?
>
> **Answer:** Paragraphs are separated **semantically**, not by sentence count. The process is explicitly defined in both dataset construction and model training:
>
> 1. **Semantic structure comes from the dataset.** During dataset generation (Section 3.1, Step 3), Gemini produces reasoning with explicit group markers, e.g.:
>
> ```
> **<Group 1> Skin Texture Anomalies**
> The skin exhibits an unnaturally smooth appearance...
>
> **<Group 2> Lighting and Shadow Inconsistencies**
> The lighting on the face is inconsistent...
> ...
> ### Answer: fake
> ```
> Each group forms one coherent paragraph (up to M = 7), ensuring that paragraphs correspond to meaningful forensic categories.
>
> 2. **DX-LLaVA learns this structure directly**
> Fine-tuning on DF-R5 teaches DX-LLaVA to: (i) produce multi-paragraph reasoning aligned with these semantic groups; (ii) preserve the `### Answer:` delimiter; and (iii) organize evidence into interpretable themes (skin, hair, lighting, geometry, etc.).
>
> 3. **PRPO uses consistent semantic parsing**
> For reward computation, paragraphs are extracted using: (i) *Group markers* (`**Group X**`), and (ii) *Final answer*: everything after `### Answer:`
>
> Fixed-length segmentation is **never** used, as it breaks semantic coherence and degrades reward quality.
>
> ---
>
> ### Question 7: Supervised PRPO vs Unsupervised PRPO
>
> **Question:** If we use ground truth labels to introduce accuracy rewards to PRPO (like GRPO), will the performance be improved?
>
> **Answer:** This is an interesting question about reward design. We conducted this ablation to compare between supervised PRPO and unsupervised PRPO. The supervised PRPO where the majority voting is replaced by the ground truth. The experiments are trained on 4 domains and tested on the held-out domain (inter-domain). The results are shown below:
>
> | Test Domain | Method               | Accuracy | Precision | Recall | F1        |
> |-------------|-----------------------|-------|--------|--------|-----------|
> | **DDIM**    | PRPO (supervised)     | 81.60 | 100.00 | 64.95  | 78.75     |
> |             | PRPO (unsupervised)   | 95.80 | 98.79  | 93.14  | **95.88** |
> | **PixArt**  | PRPO (supervised)     | 87.49 | 99.76  | 76.74  | 86.74     |
> |             | PRPO (unsupervised)   | 88.60 | 99.29  | 79.17  | **88.10** |
>
> Across both domains, unsupervised PRPO outperforms supervised PRPO, especially in recall and F1, indicating substantially better cross-domain generalization. This effect is intuitive: *label-free rewards* (VCR + PCR) encourage reasoning that is visually grounded and self-consistent, making the policy more domain-invariant. In contrast, ground-truth rewards can unintentionally overfit to domain-specific artifacts or annotation biases, hurting performance on unseen generative models. This is analogous to observations in unsupervised representation learning, where contrastive or self-supervised objectives often yield stronger cross-domain transfer than fully supervised training.

---

> > ### Comment · Reviewer_jDoc · 2025-11-24
> >
> > I thank the authors for their detailed responses. My concerns have been addressed and it's amazing to see that the unsupervised PRPO even outperforms the variant with ground truth labels. I raised my score to 8.

---

> > > ### Author Response · Authors · 2025-11-25
> > >
> > > We sincerely thank Reviewer jDoc for your insightful comments. We are glad that our clarifications with additional experiments addressed your earlier concerns. We appreciate your reassessment of the strengths of the unsupervised PRPO variant.

---

### Official Review · Reviewer_1dF7 · 2025-10-30

**Soundness:** 2
**Presentation:** 3
**Contribution:** 2
**Rating:** 4
**Confidence:** 3

**Summary:**

The paper proposes a method to improve the deepfake detection for multimodal models. To deal with limitations in existing multimodal deepfake detection, the authors introduce a new large scale dataset comprising reasoning explanations automatically generated from Gemini 2.5. The authors further modify Llava by using CLIP Convnext encoder along with auxiliary classification loss. The authors also introduce PRPO, which is a test time reinforcement learning algo using paragraph level rewards and prediction consistency reward to optimize alignment and consistency of generated reasoning.

**Strengths:**

- The paper is easy to read and well presented.
- PRPO in order to improve deepfake detection is interesting area and novel in this context.
- The paper addresses a key problem and the approach in direction of explainability has significant value.
- The authors also introduce a large scale dataset in this context, which will be useful for future exploration in this area.

**Weaknesses:**

- The proposed dataset is built on the distilled reasoning from Gemini 2.5. This can severally limit the significance of the conclusion in this paper.
- VCR relies on the semantic alignment, which could be insufficient for forensic tasks where visual consistency needs to verified at pixel level.
- The architectural modification seems to be oversold and is not a significant modification. This could be strengthened by low level analysis or some visualisations. The claim on architecture benefits are also not well justified.
- Its not clear how much computation overhead is at test-time with RL approach. The latency aspect is not well explained.
- A significant drawback of current method is around overfitting the reasoning style of Gemini 2.5 which could lead to fragile explanations.

**Questions:**

I would request the authors to provide inference time overhead for PRPO optimization. Please also look at the above weaknesses.

---

> ### Author Response · Authors · 2025-11-20
>
> We sincerely thank the reviewer for their constructive feedback and valuable suggestions. We address each concern below.
>
> ### Question 1: Dataset Quality and Gemini Dependency
>
> **Question:** The proposed dataset is built on the distilled reasoning from Gemini 2.5. This can severely limit the significance of the conclusion in this paper. There's a risk of overfitting the reasoning style of Gemini 2.5 which could lead to fragile explanations.
>
> **Answer:** This was our initial concern, and we carefully considered how to design the pipeline in a way that reduces dependence on Gemini's style and avoids fragile explanations. We note that prompt design was non-trivial, involving iterative refinement and manual review of random samples across domains. We address this concern through multiple design choices and empirical validations:
>
>
> **1. Multi-model feature discovery prevents single-model bias:**
> Our 3-step pipeline (Figure 2) deliberately uses **4 different MLLMs** in Step 1 (Gemini-2.5, GPT-4o, Qwen-2.5, LLaMA-4) to generate the initial feature set. Each model contributed K=50 features, yielding 200 diverse characteristics that were consolidated to k=74 unique features (Table 9 in Appendix). **This multi-model approach ensures the feature space reflects diverse reasoning patterns, not just Gemini's style**.
>
> **2. Structured scoring reduces hallucination:**
> Step 2 forces Gemini to score each of 74 features individually as Real/Fake/Uncertain, rather than generating free-form text. This structured approach:
> - Grounds reasoning in predefined concepts (reducing hallucination)
> - Includes self-correction: uncertain cases are refined based on STaR method [1]
> - Filtered 35k/150k samples (23%) that Gemini couldn't reliably annotate
>
> **3. PRPO moves beyond Gemini's reasoning style:**
> The core contribution of PRPO is to **refine and improve** the distilled reasoning through paragraph-level rewards:
> - VCR enforces visual grounding (reducing generic statements Gemini might generate)
> - PCR enforces self-consistency (correcting logical contradictions)
>
> Table 6 shows PRPO achieves 4.55/5.0 reasoning quality, surpassing Gemini-2.5's own 4.20/5.0. This demonstrates that PRPO-trained models **exceed the teacher's quality**, not merely mimic it.
>
> **4. Empirical validation of robustness:** To rigorously address the concern regarding potential overfitting to Gemini's reasoning style, we conducted a systematic linguistic analysis over **5,000 test examples** spanning all deepfake detection domains (randomly sampling 1,000 images per domain). For each image, we compared the generated reasoning from **DX-LLaVA** and **PRPO** with **Gemini**'s reference explanation.
>
> We applied an **n-gram overlap analysis** for $ n $ $\in$ \{1, 2, 3, 4\}. For every model-Gemini pair, we extracted all consecutive $n$-word sequences from both responses and computed the percentage of model $n$-grams that also appear in Gemini's output. We then averaged these percentages across all **5,000** pairs to obtain corpus-level similarity measures. In addition, we measured each model's reasoning diversity by counting the number of *unique visual forensic features* (distinct deepfake artifacts identified, such as "blurry edges",  "lighting inconsistencies", etc.) and *creative features* (features invented by the model beyond Gemini's vocabulary).
>
> **Table: Reasoning diversity against Gemini's reference across 5 domains**
>
> | Metric                           | DX-LLaVA | PRPO  |
> |----------------------------------|----------|-------|
> | 1-gram overlap with Gemini (%)   | 48.94%   | 48.32% |
> | 2-gram overlap with Gemini (%)   | 15.89%   | 16.05% |
> | 3-gram overlap with Gemini (%)   | 5.51%    | 5.56%  |
> | 4-gram overlap with Gemini (%)   | 2.36%    | 2.35%  |
> | Unique visual features mentioned | 550      | 1,593  |
> | Creative visual features         | 20       | 404    |
>
> The results indicate that **both models exhibit extremely low multi-gram similarity to Gemini**, under **6% for 3-grams** and approximately **2 to 3% for 4-grams**, which is far below any threshold that would suggest stylistic memorization. Moreover, **PRPO produces far richer and more diverse forensic explanations**, evidenced by nearly **three times more unique visual features** and **404 creative features** beyond Gemini's corpus. These findings provide strong evidence that the models, especially PRPO, have **learned robust and generalizable deepfake-detection reasoning** rather than overfitting to Gemini's linguistic patterns.
>
> [1] Eric Zelikman, Yuhuai Wu, Jesse Mu, and Noah D. Goodman. Star: Bootstrapping reasoning with
> reasoning. NeurIPS, 2022.

---

> ### Author Response · Authors · 2025-11-20
>
> ---
>
> ### Question 2: Visual Consistency Reward (VCR) for Forensic Tasks
>
> **Question:** VCR relies on semantic alignment, which could be insufficient for forensic tasks where visual consistency needs to be verified at pixel level.
>
> **Answer:** We appreciate this concern about the granularity of VCR. However, our design actually incorporates pixel-level forensic verification through multiple mechanisms:
>
> **1. CLIP ConvNeXT provides pixel-level embeddings.**
> Unlike CLIP ViT which outputs patch-level features (14×14 patches for 224×224 images), CLIP ConvNeXT produces **pixel-level feature maps**:
> - Stage-3 output: 10×10 spatial resolution preserving fine-grained spatial information. Each of 100 pixel embeddings captures local forensic details.
>
> **2. VCR operates on granular visual-textual alignment.**
> The VCR computation (Eq. 3) uses cosine similarity between:
> - *Text*: Keywords extracted by YAKE from each paragraph (e.g., "smooth skin", "unnatural hairline", "blurred edges")
> - *Vision*: CLIP ConvNeXT embeddings that encode pixel-level textures
>
> This is **not purely semantic**, CLIP's contrastive training ensures that visually grounded terms (e.g., "blurred") align with actual pixel-level blur patterns, not just semantic concepts.
>
> **3. Empirical validation.** Table 7 shows VCR-only achieves 88.68% F1 with 99.30% precision, demonstrating that VCR effectively identifies genuine visual artifacts (high precision) while avoiding false forensic claims. The high precision indicates VCR successfully grounds reasoning in actual visual evidence.
>
> **4. Additional experiments.**
> To further validate that VCR captures fine-grained forensic cues, we analyze its sensitivity to pixel-level changes. We randomly sample 100 images per domain and apply common frequency-domain perturbations (Gaussian noise, brightness change, Gaussian blur, contrast adjustment, JPEG compression, and rotation). For each image, we use our trained model to generate 16 responses and compute the final prediction using the averaged majority-vote accuracy (Avg Majority Vote Acc). The results are shown in Table below:
>
> **Table: Effect of Pixel-Level Perturbations on VCR Majority-Vote Accuracy**
>
> | Perturbation       | Factor / Strength          | VCR (Avg Majority Vote Acc %) | Change from Unperturbed (%) |
> |--------------------|----------------------------|-------------------------------|------------------------------|
> | **Unperturbed**    | –                          | **96.00**                     | –                            |
> | **Gaussian Noise** | sigma = 35                 | 68.25                         | −27.75                       |
> | **Brightness**     | factor = 1.4               | 95.13                         | −0.87                        |
> | **Gaussian Blur**  | kernel size = 9, sigma = 3 | 79.00                         | −17.00                       |
> | **Contrast**       | factor = 1.5               | 91.19                         | −4.81                        |
> | **JPEG Compression** | quality = 20               | 65.28                         | −30.72                       |
> | **Rotation**       | degrees = 5                | 82.75                         | −13.25                       |
>
> These results show that VCR is highly sensitive to pixel-level distortions introduced by frequency-domain perturbations. When these perturbations are applied, the generated answers become more diverse, and VCR loses consensus across the 16 responses. For example, Gaussian noise and JPEG compression lead to substantial performance drops (−27.75% and −30.72%, respectively), indicating that VCR indeed relies on fine-grained forensic consistency at the pixel level rather than solely on high-level semantic alignment.

---

> ### Author Response · Authors · 2025-11-20
>
> ---
>
> ### Question 3: Architectural Modification Justification
>
> **Question:** The architectural modification seems to be oversold and is not a significant modification. This could be strengthened by low-level analysis or some visualizations. The claim on architecture benefits are also not well justified.
>
> **Answer:** We respectfully disagree that the architectural contribution is oversold. The empirical gains are substantial, and we provide additional analysis to justify the claims:
>
> **1. Additional ablation study.** To compare the architectural impact of ViT vs. ConvNeXT, we conduct an analysis where we change the architecture of vision model in DX-LLaVA from ConvNeXT to ViT-L/14. Note that DX-LLaVA (ViT-L/14) is not the same as the original LLaVA. Here we used second-to-last block of ViT-L/14 that spatial feature map. All other set up are kept the same for fair comparision. The result is shown below:
>
> **Table: Comparison of ViT-L/14 and ConvNeXT Backbones With/Without Auxiliary Binary Loss**
>
> | Vision Backbone          | Aux. Binary Loss     | Intra-Domain F1 | Inter-Domain F1 (DDIM) |
> |--------------------------|-----------------------|-----------------|------------------------|
> | DX-LLaVA (ViT-L/14)      | without binary loss   | 96.74           | 61.21                  |
> | DX-LLaVA (ViT-L/14)      | with binary loss      | 96.25           | 67.41                  |
> | DX-LLaVA (ConvNeXT)      | without binary loss   | **99.91**       | **80.86**              |
> | DX-LLaVA (ConvNeXT)      | with binary loss      | **99.76**       | **91.54**              |
>
> The results show that ConvNeXT outperforms ViT-L/14 by a large margin in both intra-domain (+3.51% F1) and inter-domain (+24.13% F1) settings, confirming that the architectural choice is critical for deepfake detection.
>
>
> **2. Architectural difference is fundamental, not superficial:**
> The ViT vs. ConvNeXT distinction is not merely "swapping encoders", it reflects fundamentally different inductive biases:
>
> **Table: ViT vs. ConvNeXT: architectural comparison for deepfake detection**
>
> | Property | CLIP ViT-L/14 | CLIP ConvNeXT-L                 |
> |----------|--------------|---------------------------------|
> | Embedding type | Patch-level (14×14) | High-resolution conv feature map |
> | Receptive field | Global (self-attention) | Hierarchical convolution        |
> | Inductive bias | Weaker texture bias | Strong local texture bias       |
> | High-frequency sensitivity | Lower | Higher                          |
> | Parameters | 304M | 352M                            |
>
> **3. Additional visualizations:**
> We added the following analyses to the appendix for your reference:
>
> - **Saliency maps**: Saliency visualizations show that ConvNeXT focuses on forensic artifacts. These results are provided in **Figure 10, Appendix A.6**.
>
> - **Feature visualization and prediction confidence**: We use $t$-SNE visualizations to show that ConvNeXT features form clear real/fake clusters in both intra-domain and inter-domain settings, whereas ViT features exhibit poor separation and weak cross-domain generalization. Furthermore, we plot the confidence distributions and demonstrate that the ConvNeXT-based architecture produces more confident predictions (higher softmax probabilities) than ViT-based models, as shown in **Figures 11 and 12** in **Appendix A.6**.

---

> ### Author Response · Authors · 2025-11-20
>
> ---
>
> ### Question 4: Test-time Computational Overhead
>
> **Question:** It's not clear how much computation overhead is at test-time with RL approach. The latency aspect is not well explained. Please provide inference time overhead for PRPO optimization.
>
> **Answer**: To investigate the computational cost, we profile the model over the first 10 iterations and report the averaged measurements. All experiments are conducted on 8 NVIDIA H200 GPUs. The input image resolution is set to 320×320, and the maximum generation length is fixed at 1024 tokens. We vary the batch size from 8 to 32 and measure the following metrics:
>
>  **Table 1. PRPO Training Performance Across Batch Sizes**
>
> | Batch Size | Avg Iter Time (s) | Std Iter Time (s) | Avg Throughput (samples/s) | Std Throughput (samples/s) | Avg Latency (ms) | Std Latency (ms) |
> |-----------:|-------------------:|--------------------:|-----------------------------:|------------------------------:|-------------------:|-------------------:|
> | 8          | 110.7902           | 32.5974             | 0.0810                       | 0.0311                        | 13848.78           | 4074.68            |
> | 16         | 153.0948           | 45.6196             | 0.1131                       | 0.0294                        | 9568.43            | 2851.22            |
> | 32         | 237.6357           | 37.3804             | 0.1379                       | 0.0211                        | 7426.12            | 1168.14            |
>
> Across different batch sizes, PRPO exhibits a clear efficiency trend. Increasing the batch size from 8 to 32 improves throughput from 0.0810 to 0.1379 samples/s (+70%) and reduces per-sample latency from 13.8s to 7.4s (-46%), indicating substantially better hardware utilization. Although the average iteration time rises (from 110.8s to 237.6s), this is expected because each iteration processes more data; the per-sample cost consistently decreases. Variability also improves: the standard deviation (Std) of latency drops from 4074.7 ms to 1168.1 ms, showing more stable training dynamics at larger batch sizes. Larger batches offer significantly more efficient and stable PRPO training.
>
> **Table 2. Component Breakdown of PRPO Computational Cost**
>
> | Batch Size | Sampling  (s) | Sampling  (%) | Reward (s) | Reward (%) | Actor Update (s) | Actor Update (%) | Ref Policy (s) | Ref Policy (%) | Advantage (s) | Advantage (%) |
> |-----------:|------------:|------------:|-----------:|-----------:|------------------:|------------------:|----------------:|----------------:|---------------:|----------------:|
> | 8          | 95.3877     | 89.08%      | 3.4758     | 3.25%      | 6.3794            | 5.96%             | 1.8177          | 1.70%           | 0.0222         | 0.02%          |
> | 16         | 131.0687    | 89.43%      | 6.3492     | 4.33%      | 7.1801            | 4.90%             | 1.9357          | 1.32%           | 0.0257         | 0.02%          |
> | 32         | 203.7870    | 90.14%      | 11.6396    | 5.15%      | 8.3056            | 3.67%             | 2.2918          | 1.01%           | 0.0550         | 0.02%          |
>
> Across all batch sizes, the breakdown in Table 2 shows that sampling dominates the computational cost, consistently accounting for around 89-90% of total runtime. Reward computation remains lightweight (<6%), and both the actor update and reference-policy update together contribute <7% even at larger batches. The advantage computation is negligible (<0.05%). These results confirm that PRPO’s overhead is **not from the optimization or reward components**; instead, the majority of the cost comes from **generating rollouts (Sampling)**, which is inherent to any policy-gradient method. This behavior is expected and consistent with GRPO-style algorithms. Overall, PRPO remains computationally efficient, with overhead scaling smoothly as batch size increases.

---

### Official Review · Reviewer_kPnV · 2025-11-01

**Soundness:** 3
**Presentation:** 3
**Contribution:** 3
**Rating:** 6
**Confidence:** 4

**Summary:**

This paper addresses the critical challenge of deepfake detection and forensic reasoning using Multimodal Large Language Models (MLLMs). The core contribution is the introduction of a new reasoning-annotated dataset, DF-R5 (115k image-reasoning pairs), specifically designed for deepfake detection. Furthermore, the authors propose an enhanced MLLM architecture, DX-LLaVA, fine-tuned on DF-R5, and a novel test-time optimization algorithm, Paragraph-level Policy Optimization (PRPO), to align the generated forensic explanations with visual evidence. Experimental results demonstrate that the combined approach significantly outperforms existing detection baselines and general state-of-the-art MLLMs in both classification accuracy and reasoning quality.

**Strengths:**

1. The research tackles the complex issue of boundary ambiguity and unreliable reasoning in MLLM-based deepfake detection. The proposal of an MLLM fine-tuning method (DX-LLaVA) combined with a post-SFT optimization algorithm (PRPO) for forensic reasoning is a clear and novel contribution to the field.

2. The introduced dataset, DF-R5, provides a substantial volume (115k pairs) of high-quality, fine-grained reasoning annotations crucial for training MLLMs to perform reliable deepfake forensics. This dataset significantly helps in bridging the data gap for explainable deepfake detection.

3. The PRPO method demonstrates substantial and consistent performance improvements over both dedicated deepfake detection baselines (Table 4) and general SOTA MLLMs (Table 5, Table 10), validating the effectiveness of the proposed data and optimization strategy.

**Weaknesses:**

Major concerns:

1. Lack of clarity and details regarding the proposed DX-LLaVA architecture and its implementation. Section 3.2, which describes the method, abruptly introduces DX-LLaVA without a smooth transition from the preceding section, making the rationale behind its design unclear. Specifically, the paper needs explicitly state which parameters of the base LLaVA model are fine-tuned (e.g., full LLM weights, LoRA, or just the projection layer). Similarly, the replacement of the vision encoder with ConvNeXT is introduced suddenly. Furthermore, the embedding of certain experimental results (Tables 2 and 3) directly within the methodology section hinders the reader's ability to first grasp the complete method overview before evaluating its performance.

2. Inconsistency and ambiguity in the experimental setups, which complicates the assessment of performance gains. The paper needs to clearly articulate the data split. Specifically, are the experiment results shown in Table 3 based on an inter-domain or intra-domain split. This distinction is crucial for understanding and validating the reported performance improvements.

3. The ambiguity of the relationship between DX-LLaVA and PRPO, and lack of their individual contributions in the ablation analysis. While PRPO is understood to operate on the SFT-trained DX-LLaVA, the paper lacks a clear ablation study.

**Questions:**

See Weaknesses.

---

> ### Author Response · Authors · 2025-11-20
>
> We sincerely thank the reviewer for their constructive feedback and valuable suggestions. We address each concern below.
>
> ### Question 1: DX-LLaVA Architecture Details
>
> **Question:** Lack of clarity and details regarding the proposed DX-LLaVA architecture. The paper needs to explicitly state which parameters of the base LLaVA model are fine-tuned (e.g., full LLM weights, LoRA, or just the projection layer). The replacement of the vision encoder with ConvNeXT is introduced suddenly without smooth transition.
>
> **Answer:** We appreciate this feedback and will clarify the architectural details and motivation in the revision. Here are the complete details:
>
> **1. Trainable vs. Frozen components:**
>
> **Table: DX-LLaVA architecture: component-wise training configuration**
>
> | Component                        | Parameters | Trainable | Learning Rate |
> |----------------------------------|------------|-----------|---------------|
> | CLIP ConvNeXT (vision encoder)   | 196.2M     | Frozen    | -             |
> | Projection layer $W$ (2-layer MLP) | 23.1M      | ✓         | 2×10⁻⁵        |
> | Vicuna-7B (language model)       | 6.738B     | ✓         | 2×10⁻⁵        |
> | Classifier $\mathcal{C}$(·; $\phi$) (2-layer MLP) | 2.1M       | ✓         | 2×10⁻⁵        |
> | **Total trainable**              | **6.763B** | -         | -             |
>
> We employ **full fine-tuning** (rather than LoRA) for both the LLM and the projector to ensure maximum adaptation to the deepfake detection task. This choice is supported by our sufficiently large training dataset (115k samples), which allows stable full-parameter optimization. The vision encoder, however, is kept frozen to preserve its strong pretrained texture-aware representations.
>
> **2. Abrupt introduction of ConvNeXT**
>
> Actually, we planned to include an additional table comparing ViT and ConvNeXT architectures to provide a smoother transition and justify our selection. Due to space limitations, we removed that table, which made the surrounding explanation denser. In the revised version, we will improve the narrative flow and introduce ConvNeXT more clearly and coherently.
>
> ---
>
> ### Question 2: Experimental Setup Clarity
>
> **Question:** Inconsistency and ambiguity in experimental setups. The paper needs to clearly articulate the data split. Specifically, are the experiment results shown in Table 3 based on an inter-domain or intra-domain split?
>
> **Answer:** We would like to clarify the experimental setup used throughout the paper:
>
> **Table 2 (Intra-domain vs. Inter-domain):**
> This table compares two data-splitting strategies on the baseline LLaVA model:
> - **Intra-domain:** Using the full 115k samples (23.5k DDIM, 23.3k PixArt, 24.3k SD, 21.3k SiT, 23.1k StyleGAN3), we randomly split the dataset into train/validation/test with a ratio of 98%/1%/1%. All splits contain mixed samples from all five domains.
> - **Inter-domain:** A leave-one-domain-out protocol. In each fold, we train on 4 domains (e.g., DDIM, PixArt, SD, SiT) and evaluate on the held-out domain (e.g., StyleGAN3).
>
> **Table 3 (Ablation of binary loss):**
> This table uses the **inter-domain** setting to evaluate the effect of adding the binary classification loss $L_{binary}$, to the language modeling loss  $L_{lm}$
> . The large improvement (from 35.82% to 61.66% F1) indicates that joint optimization with a classification objective substantially enhances generalization to unseen domains. Indeed, the results for the $L_{lm}$-only row are identical to the last row of Table 2.
>
> We will add additional details about the experimental setups in Tables 2 and 3 to avoid any ambiguity.

---

> ### Author Response · Authors · 2025-11-20
>
> ---
>
> ### Question 3: Ablation Study
>
> **Question:** The ambiguity of the relationship between DX-LLaVA and PRPO, and lack of their individual contributions in the ablation analysis. While PRPO is understood to operate on the SFT-trained DX-LLaVA, the paper lacks a clear ablation study.
>
> **Answer:** We will clarify the relationship between DX-LLaVA and PRPO with an additional ablation table. Below, we present experiments designed to **isolate the contributions of each component**.
>
> **Table: Ablation study of the relationships between LLaVA, DX-LLaVA, and PRPO (inter-domain on task → DDIM)**
>
> | Architecture      | PRPO Applied | Accuracy    | Precision | Recall | F1     |
> |-------------------|--------------|--------|-----------|--------|--------|
> | LLaVA baseline    | ✗            | 59.40  | 98.26     | 21.90  | 35.82  |
> | LLaVA baseline    | ✓            | 62.70  | 98.72     | 29.33  | 45.23  |
> | DX-LLaVA          | ✗            | 92.60  | 99.11     | 86.43  | 92.34  |
> | DX-LLaVA          | ✓            | 95.80  | 98.79     | 93.14  | 95.88  |
>
>
> This experiment demonstrates that both **DX-LLaVA** and **PRPO** contribute substantially to the overall performance:
>
> - **DX-LLaVA** yields a large improvement in inter-domain generalization, increasing F1 from **35.82%** to **92.34%** (+56.52%) through architectural modifications.
> - **PRPO** provides an additional performance boost by **test-time optimization**, improving inter-domain F1 from **35.82%** to **45.23%** (+9.41%) on the LLaVA architecture and from **92.34%** to **95.88%** (+3.54%) on the DX-LLaVA architecture.
>
> The results confirm that DX-LLaVA addresses architectural limitations of LLaVA, while PRPO enhances reasoning quality and robustness during inference.
>
> ---
>
> ### Question 4: Tables Placement
>
> **Question:** The embedding of certain experimental results (Tables 2 and 3) directly within the methodology section hinders the reader's ability to first grasp the complete method overview before evaluating its performance.
>
> **Answer:** We will reconsider adding Tables 2 and 3 to ensure a smoother transition and clearer narrative flow. Thank you for your suggestion.

---

> > ### Comment · Reviewer_kPnV · 2025-11-26
> >
> > Thank the author for the detailed responses. Most of my concerns are addressed. Please include the above clarifications in the revised manuscript.
> >
> > Notably, if the results for the $L_{lm}$-only row are identical to the last row of Table 2, the precision of in Table 3 ($L_{lm}$-only row) is 70.40%, inconsistent with the 98.26% in Table 2 (Inter-domain row), is a typo?

---

> > > ### Author Response · Authors · 2025-11-26
> > >
> > > We thank the reviewer for taking the time to respond to our rebuttal.
> > >
> > > **"...70.40%, inconsistent with the 98.26% in Table 2..., is a typo?"**
> > >
> > > Yes, that is correct. Thank you for pointing this out. The correct value is 98.26%, and we have corrected it in the revised manuscript.
> > >
> > > **"Please include the above clarifications in the revised manuscript."**
> > >
> > > We confirm that all clarifications have been incorporated into the updated version. Specifically:
> > > - We added the DX-LLaVA architecture table (from Question 1) to Appendix A.7 (Table 12), which is referenced in Section 4.2 (Implementation Details).
> > > - We added a new paragraph titled "Intra-domain vs. Inter-domain" in Section 4.2, providing detailed explanations of the evaluation settings used in Tables 2 and 3.
> > > - We introduced a new ablation study in Section 4.7: Effectiveness of DX-LLaVA and PRPO. The table mentioned in Question 3 corresponds to Table 9 in the main manuscript, and we have included additional analysis accompanying it.
> > >
> > > If further clarification or revisions are needed, we are happy to make additional updates.

---

### Official Review · Reviewer_24DX · 2025-11-01

**Soundness:** 3
**Presentation:** 3
**Contribution:** 3
**Rating:** 6
**Confidence:** 2

**Summary:**

The paper proposes a framework to enhance both detection accuracy and interpretability of multimodal reasoning in deepfake image detection. The authors address two key gaps: the scarcity of large reasoning-annotated datasets and the misalignment between visual evidence and generated explanations by multimodal large language models (MLLMs). To tackle these, they first introduce DF-R5, a dataset of approximately 115 K images paired with high-quality reasoning annotations; second, they present DX-LLaVA, a vision-language architecture combining a CLIP ConvNeXT encoder with a language model to better capture fine-grained artifacts; and third, they develop Paragraph-level Relative Policy Optimization (PRPO), a test-time reinforcement-learning algorithm that rewards paragraph-level alignment of reasoning with visual features and internal consistency between reasoning and final decision

**Strengths:**

* The dataset contribution (DF-R5) is significant: reasoning annotations for deepfake detection are rare, and the scale (~115k) is commendable.
* The architectural component (DX-LLaVA) shows awareness of the limitation of generic vision encoders for subtle artifact detection; replacing CLIP ViT with ConvNeXT is a reasonable design choice.
* The RL component (PRPO) is novel in that it treats paragraphs as units for reward, rather than tokens or single outputs, and aligns reasoning with visual features as well as final output consistency.

**Weaknesses:**

One concern is that the proposed model has been fine-tuned specifically on deepfake data, while the compared baselines are general-purpose MLLMs such as Gemini-2.5 and GPT-4o. This raises a fairness issue: the reported performance gains might largely reflect domain-specific adaptation rather than the effectiveness of the proposed algorithm itself.

**Questions:**

A key question concerns the generality of the proposed PRPO framework. While the method is shown effective for deepfake detection, it remains unclear whether PRPO can generalize to broader vision-language reasoning tasks, such as visual question answering or visual entailment. Given that the algorithm optimizes reasoning alignment at the paragraph level, one would expect it to be applicable beyond this specific domain. I encourage the authors to clarify this point or discuss potential extensions to more general visual reasoning settings.

---

> ### Author Response · Authors · 2025-11-20
>
> We sincerely thank the reviewer for their constructive feedback and valuable suggestions. We address each concern below.
>
> ### Question 1: Fairness of Comparison
>
> **Question:** The proposed model has been fine-tuned specifically on deepfake data, while the compared baselines are general-purpose MLLMs such as Gemini-2.5 and GPT-4o. This raises a fairness issue: the reported performance gains might largely reflect domain-specific adaptation rather than the effectiveness of the proposed algorithm itself.
>
> **Answer:** We appreciate the concern. We provide clarification to ensure that our comparisons are fair and well-controlled:
>
> **1. Comparison with a 2-criteria design:** We conduct comparisons at two distinct levels:
> - **Task-specific models (Table 4)**: DX-LLaVA vs. **fine-tuned** deepfake detectors (DE-FAKE, FakeShield, UnivCLIP, SIDA). All models in this comparison are trained on our deepfake data, ensuring a fair evaluation. In this setting, DX-LLaVA outperforms the current state-of-the-art baseline (SIDA) by 2.82% on average, demonstrating the effectiveness of both our architectural design and dataset.
>
> - **General MLLMs (Table 5)**: DX-LLaVA vs. zero-shot MLLMs (GPT-4o, Gemini-2.5, Gemma-3, etc.) to illustrate the performance gap between general-purpose models and task-adapted models.
>
> **2. PRPO's contribution goes beyond domain adaptation:** Our core contribution is PRPO, which yields a +11.83% improvement over the already fine-tuned DX-LLaVA (78.08% → 89.91% in Table 4). This demonstrates that PRPO provides significant benefits beyond domain adaptation. The ablation in Table 8 further shows that PRPO outperforms other RL methods (PPO, GRPO) under **identical fine-tuning conditions**.
>
> **3. Generalizability of PRPO:** To further validate PRPO's algorithmic contribution, we perform an additional experiment by applying PRPO to different base models:
>
> **Table: PRPO applied to different base models (inter-domain F1% on task → DDIM)**
>
> | Base Model         | Without PRPO (Inter-F1) | With PRPO (Inter-F1) | Gain (↑) |
> |--------------------|--------------------------|----------------------|-----------|
> | LLaVA-1.5 (7B)     | 35.82                    | **45.23**            | +9.41    |
> | DX-LLaVA (ours)    | 92.34                    | **95.88**            | +3.54    |
>
> These results confirm that PRPO consistently enhances performance across different architectures, with notable gains in both LLaVA-1.5 (9.41%) and DX-LLaVA (3.54%). This demonstrates that PRPO's effectiveness is not limited to a specific model or domain, but rather reflects its general capability to improve vision-language reasoning.
>
> ---
>
> ### Question 2: PRPO in Broader Context
>
> **Question:** While the method is shown effective for deepfake detection, it remains unclear whether PRPO can generalize to broader vision-language reasoning tasks, such as visual question answering or visual entailment. Given that the algorithm optimizes reasoning alignment at the paragraph level, one would expect it to be applicable beyond this specific domain.
>
> **Answer**: This is also a great point that we have carefully considered. However, current datasets for broader vision-language reasoning tasks introduce several limitations. In Visual Question Answering (VQA) datasets, answers are typically short, ranging from **single words** to **a few sentences** (e.g., "yes/no", "the color of the curtains near the cat is red."), rather than multi-paragraph explanations. This structure hinders the direct application of PRPO, which is designed to optimize long-form reasoning. In addition, most VQA datasets do not provide paragraph-level reward signals (e.g., visual consistency or prediction consistency) that PRPO relies on during optimization.
>
> To the best of our knowledge, we are one of the few teams working specifically on long-form vision-language reasoning, and PRPO has demonstrated strong effectiveness on tasks requiring **structured, multi-step explanations**. Under similar conditions, PRPO can indeed be adapted to other vision-language reasoning tasks based on its core reward mechanisms:
>
> - **VCR (Visual Consistency Reward)**: Ensures that textual reasoning is grounded in the visual content; applicable to tasks requiring grounded explanations (e.g., VQA, visual entailment, image captioning).
>
> - **PCR (Prediction Consistency Reward)**: Encourages self-consistency in multi-step reasoning; applicable to tasks that benefit from coherent chains of thought (e.g., mathematical reasoning, commonsense QA).
>
> Nevertheless, fully exploring these broader applications requires substantial additional experimentation and goes beyond the scope of our current work, which focuses on deepfake detection with explanations across unseen deepfake domains. We will add a discussion in the Conclusion section to clarify PRPO's scope, limitations, and potential for future extension.

---

### Note · Program_Chairs · 2026-01-17
**Submission Desk Rejected by Program Chairs**

The following references in this submission do not refer to real documents and/or have major errors in bibliographic information:

 Xun Yi, Esther Walia, and Mohammed Babar. "Generative adversarial networks for medical image synthesis: A review." Medical Image Analysis, 51:1-18, 2019.